# Perinatal High-Salt Diet Induces Gut Microbiota Dysbiosis, Bile Acid Homeostasis Disbalance, and NAFLD in Weanling Mice Offspring

**DOI:** 10.3390/nu13072135

**Published:** 2021-06-22

**Authors:** Qing Guo, Yi Tang, Ying Li, Ziyuan Xu, Di Zhang, Jiangtao Liu, Xin Wang, Wei Xia, Shunqing Xu

**Affiliations:** Key Laboratory of Environment and Health (HUST), Ministry of Education & Ministry of Environmental Protection, and State Key Laboratory of Environmental Health (Incubation), School of Public Health, Tongji Medical College, Huazhong University of Science and Technology, Wuhan 430030, China; d201881286@hust.edu.cn (Q.G.); tangyi_66237@126.com (Y.T.); D201981388@hust.edu.cn (Y.L.); M202075392@hust.edu.cn (Z.X.); M202075514@hust.edu.cn (D.Z.); D202081571@hust.edu.cn (J.L.); D201881255@hust.edu.cn (X.W.); xust@hust.edu.cn (S.X.)

**Keywords:** high-salt diet, non-alcoholic fatty liver disease, hepatic inflammation, bile acid, gut microbiota

## Abstract

A perinatal high-salt (HS) diet was reported to elevate plasma triglycerides. This study aimed to investigate the hypothesis that a perinatal HS diet predisposed offspring to non-alcoholic fatty liver disease (NAFLD), the hepatic manifestation of abnormal lipid metabolism, and the possible mechanism. Female C57BL/6 mice were fed a control diet (0.5% NaCl) or HS diet (4% NaCl) during pregnancy and lactation and their offspring were sacrificed at weaning. The perinatal HS diet induced greater variation in fecal microbial beta-diversity (β-diversity) and increased bacteria abundance of *Proteobacteria* and *Bacteroides*. The gut microbiota dysbiosis promoted bile acid homeostasis disbalance, characterized by the accumulation of lithocholic acid (LCA) and deoxycholic acid (DCA) in feces. These alterations disturbed gut barrier by increasing the expression of tight junction protein (*Tjp*) and occludin (*Ocln*), and increased systemic lipopolysaccharide (LPS) levels and hepatic inflammatory cytokine secretion (TNF-α and IL-6) in the liver. The perinatal HS diet also inhibited hepatic expression of hepatic FXR signaling (*CYP7A1* and *FXR*), thus triggering increased hepatic expression of pro-inflammatory cytokines (*TNF-α* and *IL-6*) and hepatic lipid metabolism-associated genes (*SREBP-1c*, *FAS*, *ACC*), leading to unique characteristics of NAFLD. In conclusion, a perinatal HS diet induced NAFLD in weanling mice offspring; the possible mechanism was related to increased bacteria abundance of *Proteobacteria* and *Bacteroides*, increased levels of LCA and DCA in feces, and increased expressions of hepatic FXR signaling.

## 1. Introduction

Non-alcoholic fatty liver disease (NAFLD) has become the most common form of chronic liver disease in children and adolescents globally in recent years [1]. NAFLD in children is a major risk factor for the development of cirrhosis and hepatocellular carcinoma in adult [2,3], and it is also an early predictor of hypertension and type 2 diabetes [4,5]. Pregnancy is a critical period for fetal organ development [6], and maternal diet during pregnancy plays an important role in protecting against or exacerbating the offspring’s risk of developing NAFLD [7,8]. Emerging evidence from human and animal studies has revealed that a high-salt (HS) diet in adulthood is not only a major risk factor for hypertension [9] and cardiovascular disease [10], but also exerts effects on hypertriglyceridemia [11], oxidative stress, and inflammation [12], which are well-known pathological features of the development of NAFLD [13,14]. In addition, a previous epidemiological study in adults showed that HS diet was associated with increased frequencies of NAFLD [15]. However, whether a perinatal HS diet can induce NAFLD in offspring and the related mechanism remains unknown.

Gut microbiota dysbiosis plays an essential role in the development of metabolic diseases [16], and recent studies have also implicated gut microbiota dysbiosis as an important factor in the pathogenesis of NAFLD. Fecal microbiomes of children with NAFLD showed greater variation in fecal microbial beta-diversity (β-diversity) and higher abundance of *Proteobacteria*, which were demonstrated to lead to liver fibrosis [17]. Maternal diet during pregnancy could modulate the maternal microbiome and shape the gut microbiota of neonates [18]. Breast milk, which is affected by maternal diet, is also a critical factor in modulating offspring gut microbiota [19,20]. A previous study reported that a HS diet could increase the Firmicutes/Bacteroidetes ratio and decrease the abundance of Lactobacillus in the gut of C57BL/6J mice [21], while the changes in these gut microbiota have been reported to be associated with metabolic disease [22]. Moreover, gut microbiota can affect bile acid metabolism. Elevated lithocholic acid (LCA) and deoxycholic acid (DCA) in mice feces [23,24] by gut bacteria are well-known to increase intestinal barrier permeability [25] and decrease the expression of hepatic farnesoid X receptor (*FXR*) [26], which are early events in the development of NAFLD [27,28,29]. These findings led us to hypothesize that a perinatal HS diet may interfere with offspring gut microbiota and bile acid, and then contribute to NAFLD in mice offspring.

To the best of our knowledge, this is the first study to investigate the effect of maternal HS diet during gestation and lactation on NAFLD in weanling mice offspring. We also examined gut microbiota, bile acid metabolism, and hepatic FXR signaling in the pathogenesis of NAFLD in mice offspring.

## 2. Materials and Methods

### 2.1. Animal Model

All experimental procedures were approved by the Institutional Animal Care and Use Committee of Huazhong University of Science and Technology (Wuhan, China) with a certificate of application for the Use of Animals dated 1 January 2019 (approval ID: SCXK 2015-0001). Male and female C57BL/6 mice were purchased from Chang Sheng biotechnology (Liaoning, China) and housed under standard environmental conditions (12 h light/dark cycle, 50–70% humidity, and 20–25 °C). Food and water were provided ad libitum. Female C57BL/6 mice were divided into two groups: (1) ND group: fed a normal diet (ND) (0.5% NaCl) and tap water ad libitum; (2) HS group: fed an HS diet (4% NaCl) diet and tap water containing 1% NaCl ad libitum. The treatment was performed from 1 week before mating until weaning of the offspring at 3 weeks of age. All diets were purchased from Beijing Huafukang Bioscience (Beijing, China). The composition and formulation of the mouse diet are detailed in Appendix A. One male C57BL/6 mouse was paired with per two females to produce offspring. On the 14th day of gestation, pregnant mice were caged individually for delivery, and old padding was used with the cages to relieve pressure on pregnant mice. Within 5 days after delivery, the offsprings were randomly adjusted to 8 mice without hand contact with other mice. Offspring were weaned at 3 weeks of age, and body weights were recorded at 7:00 a.m. The offspring mice were fasted from 7:30 a.m. to 12:30 p.m. and stool particles were collected. After the offspring were euthanized by anesthesia with 10 mg/mL pentobarbital sodium, plasma, colon, and liver were collected and stored at −80 °C until biochemical analysis. Six mice per group was used for the following assays.

### 2.2. Plasma and Liver Index Test

Triglyceride (TG) and total cholesterol (TC) in the liver and plasma were measured using commercial kits (Nanjing Jiancheng Bioengineering Institute, China) that were adapted to 96-well microtiter plates. All the procedures were conducted according to the manufacturer’s instructions. Absorbance was then measured at the wavelength of 450 nm by a microplate reader (SYNGEN, BioTek Instruments, Winooski, VT, USA). Plasma endotoxin (lipopolysaccharide, LPS) concentration was measured using commercial kits (Nanjing Jiancheng Bioengineering Institute, China). Absorbance was then measured at the wavelength of 580 nm by a microplate reader (SYNGEN, BioTek Instruments, Winooski, VT, USA).

### 2.3. Liver Histological Assay

Liver tissues were dissected on ice, fixed in 10% formalin solution, embedded in paraffin, and sectioned into 5–7 μm. The slices were stained with hematoxylin and eosin (H&E) or Masson’s trichrome reagent, and then observed under a light microscope (200×) (Olympus IX71, Tokyo, Japan). Histological analysis was quantified by the modified Pediatric NAFLD Histological Score (PNHS) [30]. Histological features were scored on a scale of 0–3 on steatosis, 0–3 on lobular inflammation, and 0–2 on portal inflammation. Ballooning and fibrosis were not included in the scoring, since they are characteristic features of NASH, which was not seen in this model.

### 2.4. 16S rRNA Gene Sequence Analysis

The mice were placed in a sterile mouse cage lined with sterile filter paper, and stool samples were collected using sterile tweezers immediately after defecation. The filter paper was changed for each mouse sample. About 0.2 g (3~5 grains) were collected from each mouse into sterile tubes and immediately stored at −80 °C until analysis. Stool DNA was extracted using the QIAamp Fast DNA Stool Mini Kit (Qiagen, CA, USA) according to manufacturer’s instructions. Following DNA extraction, the V3-V4 region of the bacterial 16S rRNA gene was amplified by PCR using 338F (5′-ACTCCTACGGGAGGCAGCAG-3′) and 806R (5′-GGACTACHVGGGTWTCTAAT-3′), and then sequenced using the MiSeq platform (Illumina, San Diego, CA, USA). The bioinformatic-analysis readings were clustered into species-level operational taxonomic units (OTUs) at the 97% identity level. To attenuate the effect of spurious sequences, OTUs with less than 0.005% of the total number of sequences were removed [31]. After filtering, an average of 34,352 readings per sample were obtained. Calculations of within-community diversity (α-diversity), between-community diversity (β-diversity), relative abundance taxonomic summaries, and the different statistical analyses were preformed using QIIME. Principal coordinate analysis (PCOA) on unweighted UniFrac metrics between the HD and ND groups was tested using the ANOSIM statistical method.

### 2.5. Mouse Enzyme-Linked Immunosorbent Assay (ELISA) Test

Tumor necrosis factor (TNF)-α and interleukin (IL)-6 in the liver were measured using a mouse enzyme-linked immunosorbent assay (ELISA) kit (MULTISCIENCES, Hangzhou, China). All the procedures were conducted according to the manufacturer’s instructions. Absorbance was then measured at the wavelength of 450 nm by a microplate reader (SYNGEN, BioTek Instruments, Winooski, VT, USA).

### 2.6. Bile Acid Metabolism Analyses

Freeze-dried liver and fecal samples were weighed, 5 mg samples were placed into 50 μL of 100 ng/mL internal standard (taurochenodeoxycholic acid-d5, cholic acid-d4, gluconic acid-d4, ursodexycholic acid-d4, glycochenodeoxycholic acid-d4, glycocholic acid-d4, lithocholic acid-d4, ursodeoxycholic acid-d4, chenodeoxycholic acid-d4, eoxycholic acid-d4, and taurocholic acid-d5) and 250 μL of 0.1 M NaOH. First, the mixture was reacted in an oven at 80 °C for 1 h. Next, the samples were placed into 700 μL of 2% acetonitrile solution, and then were centrifuged at 13,000 rpm for 5 min at 4 °C to collect the supernatant. After the SPE column was activated by 1 mL of methanol and equilibrated by 2 mL of 2% acetonitrile solution, 900 μL of supernatant were loaded on. The column was then washed by 3 mL of 2% acetonitrile, 3 mL of n-hexane, 3 mL of 2% acetonitrile solution, and 4 mL of methanol solution, separately. Lastly, the eluents were collected, blown dry with nitrogen, and reconstituted with 150 μL methanol. The bile acids from the liver and feces were quantitatively measured by AB Sciex API 4000 ™ LC/MS (Applied Biosystems, Mississauga, ON, Canada).

### 2.7. qPCR RNA

RNA was extracted from liver and colon using RNAiso Plus reagent (Takara, Kusatsu, Japan) according to the manufacturer’s instructions. RNA samples were reverse-transcribed into cDNA with a HiScript^®^ III RT SuperMix for qPCR kit (Vazyme Biotech co.ltd, Nanjing, China). The cDNA samples were amplified by qPCR with a Vazyme SYBR qPCR kit (Vazyme Biotech co.ltd, China). Each primer set was verified by analysis of their melt curves, and the assays were performed in a 7900HT Fast Real-Time PCR System (Applied Biosystems, Waltham, MA, USA). The relative gene expression was calculated by the 2−ΔΔCt method with *Gapdh* as the internal control. Although studies recommended that it is better to use more than one reference gene for PCR [32], we compared multiple reference genes, and *Gapdh* is the most stable. The sequence of the primers used for qRT-PCR experiments is provided in Appendix A. Data were analyzed with the 2−ΔΔCt method with *Gapdh* gene expression used as a reference [22].

### 2.8. Statistics

Data analysis was performed using GraphPad Prism (v. 8.0). An F-test was used to test the normal distribution. Student’s *t*-test was used to analyze for normally distributed data, and a Mann–Whitney U test was used for not normally distributed data. *p* < 0.05 was considered as a significant difference and data are expressed as mean ± SEM.

## 3. Results

### 3.1. Perinatal HS Diet Induced NAFLD in the Weanling Offspring

Compared to the offspring with perinatal ND diet the body weights of the weanling offspring with perinatal HS diet were reduced significantly (Figure 1A), while liver weight/body weight of the offspring with perinatal HS diet displayed an increasing trend without statistical difference (Figure 1B). The perinatal HS diet did not alter plasma TC levels (Figure 1D); however, levels of plasma TG, hepatic TG, and TC were increased in the offspring with perinatal HS diet (Figure 1C,E,F). In addition, H&E staining of liver tissue showed unique histologic features of NAFLD in the HS offspring, characterized by greater periportal leukocyte infiltration (Figure 1G). Masson staining of liver tissue showed slight fibrosis around the portal vein, while the fibrosis was not significantly different between the two groups (Figure 1H). The modified PNHS, which was used for assessing histology change in NAFLD, was significantly higher in livers of the offspring with perinatal HS diet (Figure 1I). Overall, these results showed that perinatal HS diet caused NAFLD in the weanling offspring.

### 3.2. Perinatal HS Diet Altered Hepatic mRNA Expression of FXR Signaling in the Weaning Offspring

Hepatic mRNA expression of FXR signaling, which plays an important role in the development of NAFLD, was investigated. The mRNA expressions of *Fxr* were significantly decreased in the livers of offspring with perinatal HS diet (Figure 2). The expression of hepatic small heterodimer partner (*Shp*), the downstream gene of FXR signaling, was also significantly decreased, while the expression of sterol receptor element-binding protein-1c (*SREBP-1c*), fatty acid synthetase (*FAS*), acetyl CoA carboxylase (*ACC*), *TNF-α*, and *IL-6* were increased in the livers of offspring with perinatal HS diet (Figure 2).

### 3.3. Perinatal HS Diet Induced Gut Microbiome Disorders in the Weanling Offspring

Fecal microbial β-diversity showed a marked difference between offspring with perinatal HS and ND diets, based on the phylogenetic distance metric unweighted UniFrac (Figure 3A). A perinatal HS diet also led to a decreasing trend in the abundance of *Verrucomicrobia* and *Actinobacteria* phyla, and a significant increase in the abundance of *Proteobacteria* (Figure 3B,C). Moreover, a perinatal HS diet also led to an increasing trend in the abundance of *Parabacteroides* and *Alloprevotella* genera, and a significant increase in the abundance of *Bacteroides* (Figure 3D,E).

### 3.4. Perinatal HS Diet Altered Bile Acid and Gut Permeability in the Weanling Offspring

Fecal bile acid contents of LCA and DCA were significantly increased in the offspring with perinatal HS diet compared to those offspring with perinatal ND (Figure 4A). The mRNA expression levels of tight junction protein 1 (*Tjp*) and occludin (*Ocln*) in the colon were significantly decreased and the concentration of LPS was significantly increased in the offspring with perinatal HS diet (Figure 4B,C), indicating that a perinatal HS diet alters gut permeability. Consistent with the increased levels of LCA and DCA in the feces, the levels of pro-inflammatory cytokines (TNF-α and IL-6) were significantly increased in the livers of offspring with perinatal HS diet, while the mRNA expression of cholesterol 7α-hydroxylase (*CYP7A1*), which is the ratelimiting enzyme in bile acid biosynthesis, was significantly decreased in the livers of offspring with perinatal HS diet (Figure 4D,F). Furthermore, the bile acid levels of α-muricholic acid (α-MCA) and β-muricholic acid (β-MCA) in the livers were also significantly decreased in the weanling offspring fed perinatal HS diet (Figure 4G).

## 4. Discussion

Perinatal diet is of critical importance for fetal development [33]. A perinatal HS diet was reported to increase plasma TG levels in mice offspring [11], which is an important indicator in the pathology of NAFLD in children [13]. In this study, we demonstrated that a perinatal HS diet could induce NAFLD in offspring. We also found that the perinatal HS diet increased the abundance of gut *Proteobacteria* and *Bacteroides*, and the concentration of DCA and LCA. These alterations disturbed the gut barrier by increasing expression of *Tjp* and *Ocln* [25], and increased levels of LPS and hepatic inflammatory cytokine secretion (TNF-α and IL-6) in the liver [34]. These changes inhibited the expression of hepatic FXR signaling (CYP7A1 and FXR), thus triggering increased hepatic expression of pro-inflammatory cytokines (*TNF-α* and *IL-6*) [35] and hepatic lipid metabolism associated genes (*SREBP-1c*, *FAS*, and *ACC*) [24], leading to the unique characteristics of NAFLD.

The earlier presentation of NAFLD in children characterized mainly by the presence of a predominant portal-based injury, including portal inflammation with no or mild fibrosis [2]. In this study, a perinatal HS diet led to increased hepatic periportal leukocyte infiltration. Although histologic change in the liver did not show obvious lipid deformation, the biochemical indicators of pediatric NAFLD patients appeared abnormal, including hyperlipidemia [3]. The reason for this phenomenon is the lack of accurate methods to identify liver steatosis. When the total area of steatosis is less than 20%, the histologic method for identifying liver steatosis will generate a false negative [4]. Similarly, we did not find obvious TG accumulation in tissue by H&E and Masson staining, but the levels of hepatic TG increased significantly. The modified pediatric NAFLD histological score (PNHS) [30] is a recently developed method used for assessing histology change of NAFLD to improve accuracy. Histological features were scored on a scale of 0–3 on steatosis, 0–3 on lobular inflammation, and 0–2 on portal inflammation. The PNHS method has also been used in other animal experiments, and it was proved that it can identify NAFLD-like lesions in weaned mice [22]. We observed PNHS was significantly higher in the livers of the offspring with perinatal HS diet. Accumulating epidemiological evidence suggests that a small body size during infancy and childhood is associated with a higher risk of developing NAFLD [36,37]. Moreover, dyslipidemia is also observed in children with NAFLD [38]. In accordance with the features of NAFLD in children, we found that perinatal HS diet led to decreased body weights and increased levels of plasma TG in the offspring. Consistent with our study, Clare et al. reported that Dawley rats fed a 4% salt diet during pregnancy and lactation induced lower body weights and higher level of plasma TG in offspring [11].

Hepatic FXR signaling plays an important role in the development of NAFLD [39]. Hepatic FXR is a bile acid receptor and can modulate lipid homeostasis via SHP. SHP is a FXR target gene and promoter-specific repressor. Downregulated expression of *Fxr* can repress *Shp* expressions, which in turn facilitates the expression of *SREBP-1c* [40]. SREBP-1c as a key transcription factor plays an important role in the regulation of the expression of lipogenic genes, including *ACC-1* and *FAS* [6]. Several studies have proved that the overexpression of *SREBP-1c, FAS,* and *ACC* is associated with accelerated hepatic lipogenesis in NAFLD animal models [41,42], while downregulated mRNA expression of these genes can protect development of liver steatosis [43,44]. Moreover, a study reported that downregulated expression of *Fxr* could repress *Shp* expression, which in turn could facilitate the expression of *SREBP-1c*, *ACC,* and *FAS*, and then result in the development of NAFLD [24]. Therefore, the upregulated expression of *SREBP-1c*, *FAS,* and *ACC* observed in our study suggested that perinatal HS diet induce hepatic lipogenesis and lipid accumulation. The decreased expression of *FXR* also increased the expression of *TNF-α* and *IL-6* [35]. Hepatic lipid synthesis and inflammation are crucial mechanisms in the pathogenesis of NAFLD [45].

Substantial studies have indicated that gut microbiota dysbiosis contributes to hepatic lipid synthesis and inflammation in NAFLD [46,47]. *Proteobacteria* can increase hepatic inflammation [48] and are also a marker for gut microbiota dysbiosis and metabolic diseases in human and mice models [49]. *Bacteroides* are salt-tolerant bacteria [50] and are associated with liver injury in human studies [51,52]. Animal studies also showed that the abundance of *Proteobacteria* and *Bacteroides* are higher in NAFLD [53,54]. Therefore, the increased abundance of *Proteobacteria* and *Bacteroides* may suggest liver damage and increased risk for NAFLD. To determine whether changes in lipid metabolism were induced by gut microbiota dysbiosis, 16S rRNA analysis in our study was performed. We found perinatal HS diets increased the abundance of gut *Proteobacteria* and *Bacteroides* in weanling mice offspring. The results are consistent with previous studies that HS diets induce increases in the abundance of *Proteobacteria* and *Bacteroides* [53,55].

Gut microbiota dysbiosis and its influence on bile acids were recently considered [56]. It is well known that gut microbes produce enzymes that convert primary bile acids into secondary bile acids in the intestines. *Bacteroides* have bile salt hydrolase activity, which can metabolize primary bile acids into secondary bile acids, such as DCA and LCA [23], which are cytotoxic. DCA and LCA can potentially increase gut permeability and increase the exposure of the liver to gut-derived toxins, which can promote hepatic inflammation and liver steatosis in animal models of NAFLD [57,58]. Previous studies have shown that increased abundance of *Proteobacteria* and *Bacteroides* led to elevated levels of DCA and LCA, which consequently increased levels of TNF-α and IL-6, and inhibited the expression of *CYP7A1* in the livers of mice [59,60,61]. Some studies have shown that HS intake induced gut dysbiosis and disrupted gut barrier integrity, and that antibiotics could restore HS-diet-induced gut leakiness [53,62,63]. Furthermore, the levels of *Proteobacteria*, *Bacteroides*, DCA, LCA, TNF-α, and IL-6 in mice could be reversed by antibiotic treatment [64]. In the classic pathway of bile acid biosynthesis, *CYP7A1* determines the overall rate of bile acid production and synthesizes two primary bile acids: cholic acid and chenodeoxycholic acid (CDCA) [23]. In mice, the majority of CDCA is converted to α-MCA and β-MCA, which are both agonistic with FXR [65]. Overexpression of *Cyp7a1* can activate expression of *Fxr*, whereas inhibition of the expression of *Cyp7a1* can reduce bile acid synthesis and decrease the expression of *Fxr* [24,66]. Owing to the decrease in the expression of *Cyp7a1* in our study, we tested hepatic main bile acid. Consistent with these studies, we found that increased DCA and LCA led to gut permeability, increased the levels of hepatic TNF-α and IL-6, and decreased the expression of hepatic *CYP7A1* and the contents of α-MCA and β-MCA. Based on these reasons, we think that gut dysbiosis is an important link of perinatal HS diet to the progression of NAFLD in the offspring. Future studies need to be confirmed by antibiotic experiments.

## 5. Conclusions

In conclusion, this study revealed that a perinatal HS diet contributed to NAFLD in weanling offspring. Taken together, the perinatal HS diet increased the abundance of *Proteobacteria* and *Bacteroides* and elevated the fecal LCA and DCA levels, thereby destroying the gut barrier, increasing the levels of hepatic TNF and IL-6, and inhibiting the expression of *CYP7A1* and *Fxr*, which induced hepatic inflammation and lipid accumulation, potentially contributing to the progression of NAFLD in the weanling offspring. The findings of this study expand our current knowledge of the effect of perinatal HS intake on NAFLD of weanling offspring and the role of gut microbiota on NAFLD of weanling offspring. Further studies are needed to explore whether treatment with probiotics via modulation bile acids can improve disease progress in NAFLD of weanling offspring.

## Figures and Tables

**Figure 1 nutrients-13-02135-f001:**
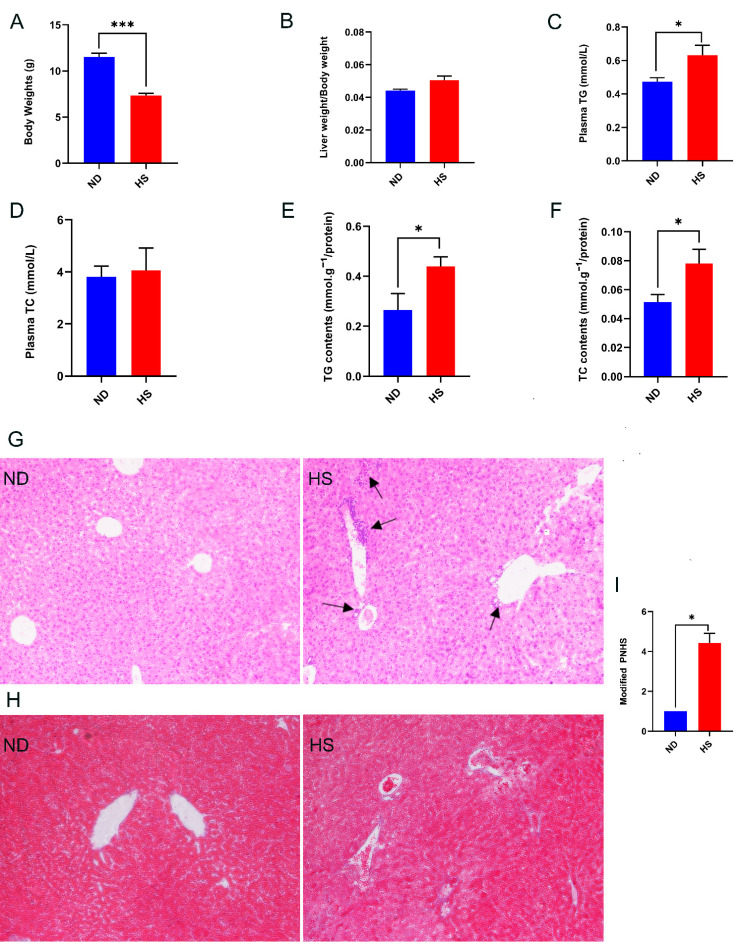
Effect of a perinatal high-salt (HS) diet on offspring growth, plasma triglyceride (TG), hepatic TG, hepatic total cholesterol (TC), and histological features of liver at 3 weeks of age compared to normal diet (ND). (**A**) Body weight. (**B**) Liver weight/body weight. (**C**) Levels of TG in plasma. (**D**) Levels of TC in plasma. (**E**) Levels of TG in liver. (**F**) Levels of TC in liver. (**G**) Representative photomicrographs of H&E staining in the liver (200×); black arrows indicate infiltrating lymphocytes. (**H**) Representative photomicrographs of Masson staining in the liver (200×). (**I**) Modified pediatric non-alcoholic fatty liver disease (NAFLD) histological score (PNHS) for offspring of perinatal HS diet and offspring of perinatal ND diet. *n* = 3 offspring of perinatal ND diet and *n* = 7 for offspring of perinatal HS diet. Data are presented as mean ± SEM. * *p* < 0.05; *** *p* < 0.001.

**Figure 2 nutrients-13-02135-f002:**
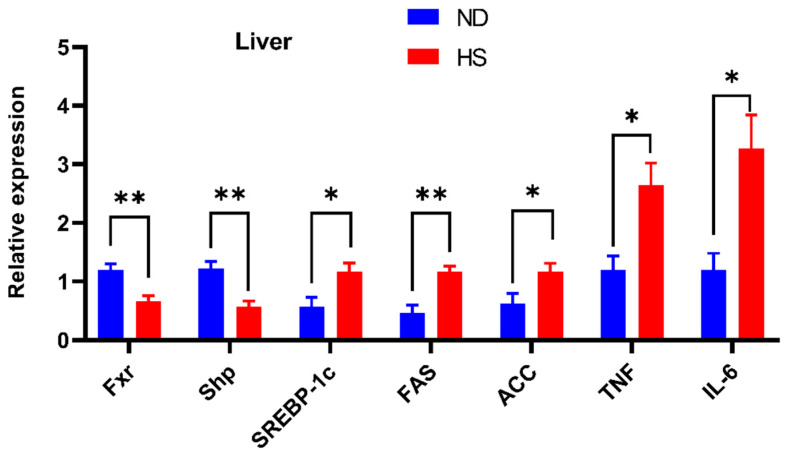
Effect of perinatal high-salt (HS) diet on offspring hepatic mRNA expression associated with farnesoid X receptor (FXR) signaling. Relative mRNA expression of sterol receptor element-binding protein-1c *(SREBP-1c),* fatty acid synthetase *(FAS)*, acetyl CoA carboxylase *(ACC)*, tumor necrosis factor *(TNF)-α,* and interleukin *(IL)-6* in the liver. The expressions were normalized to *Gapdh*. * *p* < 0.05; ** *p* < 0.01. Note: ND, normal diet.

**Figure 3 nutrients-13-02135-f003:**
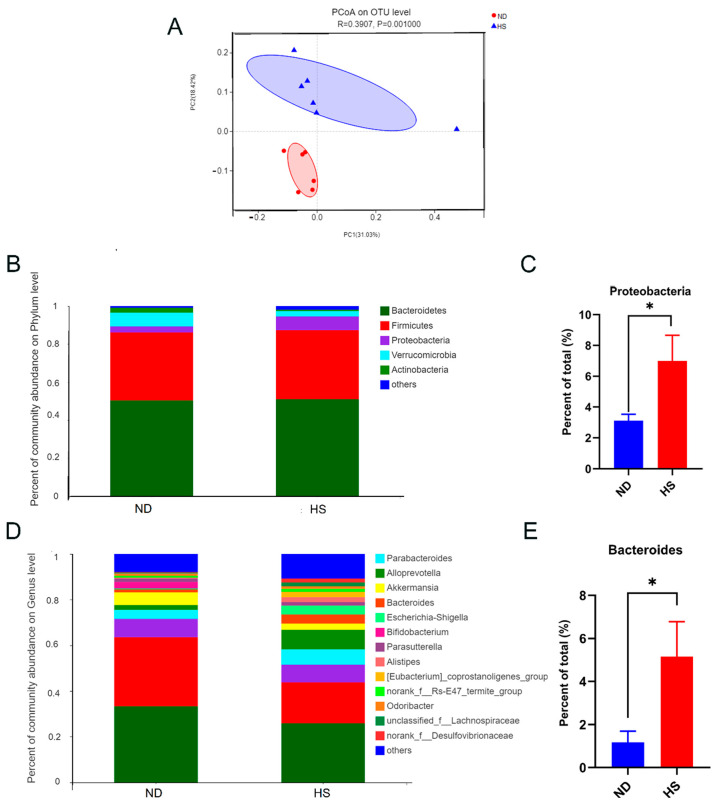
Effect of perinatal high-salt (HS) diet on gut microbiota of weanling offspring. (**A**) Principal coordinates analysis (PCoA) of unweighted Unifrac distances on the operational taxonomic unit (OTU) level. Each point represents one sample. Red dots represent normal diet (ND) samples, and blue triangles represent HS samples. Closer sample points indicate higher similarity in the species composition of the two samples. R-value represents inter-group differences and intra-group differences. A higher R-value indicates a greater difference between groups than the intragroup difference. (**B**) Relative abundance of bacterial phyla. (**C**) Relative abundances of *Proteobacteria*. (**D**) Relative abundance of bacterial genera. (**E**) Relative abundances of *Bacteroides*. * *p* < 0.05 was considered a significant diffidence between the two groups.

**Figure 4 nutrients-13-02135-f004:**
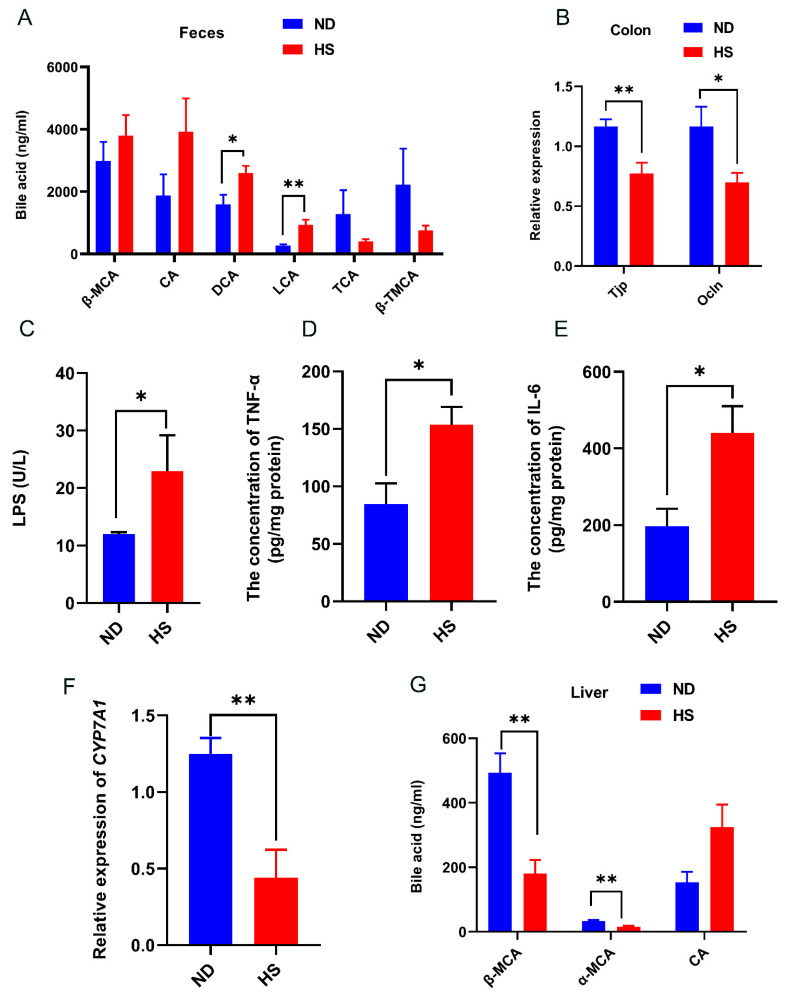
Effect of perinatal high-salt (HS) diet on offspring bile acid and gut permeability. (**A**) Levels of bile acid in feces. (**B**) Relative mRNA expression of tight junction protein (*Tjp*) and occludin (*Ocln*) in the colon, normalized to *Gapdh*. (**C**) Plasma concentration of endotoxin (LPS). (**D**,**E**) Levels of TNF-α and IL-6. (**F**) Relative mRNA expression of *CYP7A1* in the liver, normalized to *Gapdh*. * indicates *p* < 0.05; ** *p* < 0.01. (**G**) Levels of bile acid in liver. Note: ND, normal diet; α-MCA, α-muricholic acid; β-MCA, β-muricholic acid; DCA, deoxycholic acid; LCA, lithocholic acid; TCA, taurocholic acid; β-TMCA, taurosulfur-β-rocholic acid.

## Data Availability

The data presented in this study are available on request from the corresponding author.

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
