# Peer review of "Perinatal High-Salt Diet Induces Gut Microbiota Dysbiosis, Bile Acid Homeostasis Disbalance, and NAFLD in Weanling Mice Offspring"

_nutrients, 2021, doi:10.3390/nu13072135_

Round 1
Reviewer 1 Report
In this study, the authors claimed that a perinatal high-salt diet (HS) induced hepatic steatosis was associated with gut microbiota dysbiosis, bile acid homeostasis disbalance, intestinal tight junction disruption, and liver inflammation in weanling offsprings. The study is well written and easy to follow. I have few comments that may help the authors improve their manuscript.
1) Authors demonstrated that perinatal HS lead to hepatic lipid accumulation in weanling offsprings. As shown in Fig. 1E, the hepatic TG levels were significantly increased by the HS diet. However,the HE and Masson staining results do not show a corresponding TG accumulation in the liver. Furthermore,although B1 and B2 in Fig. 1H were the results of two independent samples in the same group,the difference between B1 and B2 was obvious.
2)The gut microbiota results seem too simple, authors may consider adding the total bacteria levels and bar graph to display more detailed changes at phylum and family levels.
3) Although microbiota changes were associated with tight junction disruption, whether dysbiosis plays a detrimental role in HF-induced tight junction disruption was still unclear. authors may consider using antibiotics or probiotics to prove this point.
4) Hepatic proinflammatory response is associated with tight junction disruption. Authors should measure serum or plasma endotoxins levels in mice.
5) The results obtained by the author are independent phenomena, and the inner connection of these results should be discussed more.
Author Response
I want to express my deep thanks to the reviewer’ suggestions and comments to improve the paper quality.
According to all comments from the reviewer, we revised the manuscript carefully. The changes are red-colored in the manuscript and the detailed responses to the comments are listed as the following.
Point 1: Authors demonstrated that perinatal HS lead to hepatic lipid accumulation in weanling offsprings. As shown in Fig. 1E, the hepatic TG levels were significantly increased by the HS diet. However,the HE and Masson staining results do not show a corresponding TG accumulation in the liver. Furthermore,although B1 and B2 in Fig. 1H were the results of two independent samples in the same group,the difference between B1 and B2 was obvious. 

Response 1: We agree with the HE and Masson staining results did not show a corresponding TG accumulation in the liver. Although histologic of the liver did not show obvious lipid deformation, the biochemical indicators of pediatric NAFLD patients appeared abnormal, including hyperlipidemia. The reason for this phenomenon is the lack of accurate methods to identify liver steatosis. When the total area of steatosis is less than 20%, histologic of the liver to prove liver steatosis will be inaccurate and false positive, and it is important to make the diagnosis. Similarly, we did not find obvious TG accumulation in tissue HE and Masson staining, but measured hepatic TG increased significantly. The corresponding description were added in lines 229-236 of page 9.
B1 and B2 in Fig. 1H show the structural damage of the portal vein. They were different, because some portal vein was surrounded by mild fibrosis between them.
Point 2: The gut microbiota results seem too simple, authors may consider adding the total bacteria levels and bar graph to display more detailed changes at phylum and family levels.
Response 2: The figures were added accordingly in Figure 3 B-3 E. Also, the corresponding description were added in lines 190-191 and 196-198 of page 7, respectively.
Point 3: Although microbiota changes were associated with tight junction disruption, whether dysbiosis plays a detrimental role in HF-induced tight junction disruption was still unclear. authors may consider using antibiotics or probiotics to prove this point.
Response 3: Thanks the reviewer for the great comments. Some studies have shown that HS intake can induce gut dysbiosis and disrupt gut barrier integrity, and antibiotics can restore HS intake-induced gut leakiness. Furthermore, antibiotic treatment can reverse the levels of Proteobacteria, Bacteroides, DCA, LCA, TNF and IL-6 in mice. Consistent with these studies, we found that perinatal HS diet increased the abundance of Proteobacteria and Bacteroides, elevated offspring faecal LCA and DCA levels and interfere bile acid homeostasis. Based on these reasons, we believe that gut dysbiosis is an important link that perinatal high HS induced the in the progression of NAFLD of the offspring. However, due to scientific research funding issues, we in this study did not complete the antibiotic experiment. We added the above discussion to the discussion of the article and pointed out that future studies need to be confirmed by antibiotic experiments.
Point 4: Hepatic proinflammatory response is associated with tight junction disruption. Authors should measure serum or plasma endotoxins levels in mice.
Response 4: Thanks for the good suggestions. The plasma endotoxins levels were added accordingly in Figure 4 C. Also, the corresponding description were added in lines 88-92 of page 2 and 211-212 and 215 of page 8, respectively.
Point 5: The results obtained by the author are independent phenomena, and the inner connection of these results should be discussed more.
Response 5: We agree with the reviewer. Accordingly, related revisions are made at line 245-250, 259-260 and 273-275 of page 9 and 281-282 of page 10, respectively.
SREBP-1c as a regulators of lipid homeostasis plays an important role in the regulation of lipid homeostasis [37] and hepatic FXR as a bile acid receptor can modulate SREBP-1c via SHP to control the expression of genes involved in lipogenesis, such as ACC and FAS [38]. Study has reported that down-regulated expression of Fxr can repress Shp expression, which in turn facilitates the expression of SREBP-1c, ACC and FAS, and then results in the development of NAFLD [22]. ( 245-250 of page 9)
To determine whether changes lipid metabolism was induced by gut microbiota dysbiosis, 16S rRNA analysis in our study was performed. (259-260 of page 9)
Some studies have shown that HS intake can induce gut dysbiosis and disrupt gut barrier integrity, and antibiotics can restore HS intake-induced gut leakiness [53-55]. (273-275 of page 9)
Owing to the decrease the expression of Fxr and increase the abundance of Proteobacteria and Bacteroides in our study (281-282 of page 10)

Reviewer 2 Report
Here the authors showed the possible implication of high salt (HS) diet in the offspring predisposition to NAFLD development in mice. They fed female pregnant mice with an HS diet and they observed differences in microbiota dysbiosis in offspring, which promotes bile acid homeostasis imbalance, inhibition of farnesoid X receptor, and finally lead to NAFLD.
In my opinion, the study is very well conducted and structured. The abstract is clearly exposed, the introduction provides all important information, the materials and methods are adequately described and results are well presented and then discussed. I have only a suggestion regarding the RT-PCR assay. I think that when a real-time PCR is performed it is better to use more than only one reference gene as normalizers. Currently, the guidelines to follow to perform a RT-PCR analysis recommend the use of more than one reference gene (https://doi.org/10.1373/clinchem.2008.112797). Did the authors try to use more than one? I think that it is important to validate your result.
Lastly, I listed below some oversights in the text.
- On page 3, in line 106, reference 28 is out of brackets.
- On page 4, in figure 1, graph D, the normal diet is indicated as N and not ND, please correct it.
- On page 4, in the caption of figure 1, line 155, “plasma triglyceride (TG)” is repeated twice, please remove one of the wording.
- On page 7, in line 231, the word “expressions” should be written without the final “s”, please remove it.
- On page 7, in line 240, the word “injure” should be corrected in “injury”.
- On page 7, in line 258, there is an extra space between “a“ and “MCA”, please remove it.
- I think that the quality of all figures should be increased.
Author Response
I want to express my deep thanks to the reviewers’ suggestions and comments to improve the paper quality.
According to all comments from the reviewer, we revised the manuscript carefully. The changes are red-colored in the manuscript and the detailed responses to the comments are listed as the following. Please see the attachment.
Point 1: I have only a suggestion regarding the RT-PCR assay. I think that when a real-time PCR is performed it is better to use more than only one reference gene as normalizers. Currently, the guidelines to follow to perform a RT-PCR analysis recommend the use of more than one reference gene (https://doi.org/10.1373/clinchem.2008.112797). Did the authors try to use more than one? I think that it is important to validate your result.

Response 1: We agree with the reviewer that it is better to use more than only reference gene in RT-PCR analysis. In this study, we used different sample to compare Gapdh, β-actin and HPRT reference gene and finally determined a stable Gapdh reference gene as the normalizers. In our study, the CT values of reference gene Gapdh was no significant difference and the annealing temperature of Gapdh was consistent at different times between different samples, and there was only one peak of dissolution curve for all samples. Furthermore, an article published on Nature Communication in 2018 also used the Gapdh gene of the same sequence as a reference gene for the liver of the NAFLD mouse model at weaned offspring and only this one gene was used as a reference gene. All these findings proved that the Gapdh gene is stably expressed, their abundances show strong correlation with the total amounts of mRNA present in the samples, and can control for variations in extraction yield, reverse-transcription yield, and efficiency of amplification. Also, the corresponding description were added in lines 140-142 of page 3and 143-144 of page 4, respectively.
Point 2: On page 3, in line 106, reference 28 is out of brackets.
Response 2: Thanks. Brackets have been added at line 111 of page 3.
Point 3: On page 4, in figure 1, graph D, the normal diet is indicated as N and not ND, please correct it.
Response 3: Thanks. The figures were revised accordingly in figure 1D at line 160-161 of page 5.
Point 4: in the caption of figure 1, line 155, “plasma triglyceride (TG)” is repeated twice, please remove one of the wording.
Response 4: Thanks. The one of “plasma triglyceride (TG)” has been removed at line 162 of page 5.
Point 5: On page 7, in line 231, the word “expressions” should be written without the final “s”, please remove it.
Response 5: Thanks. The final “s” has been removed at line 249 of page 9.
Point 6: On page 7, in line 240, the word “injure” should be corrected in “injury”.
Response 6: Thanks. The word “injure” have been corrected in “injury” at line 257 of page 9.
Point 7: On page 7, in line 258, there is an extra space between “a“ and “MCA”, please remove it.
Response 7: Thanks. The extra space has been removed at line 279 of page 9.
Point 8: I think that the quality of all figures should be increased.
Response 8: Thanks for the good suggestions. The figures were revised accordingly. The pixels of the picture have been increased from the original 600 pixels to 1200 pixels.
Please see the attachment for the revised version

Reviewer 3 Report
Authors investigated the effect of a maternal high salt diet during gestation and lactation of C57BL/6 wildtype mice on the development of NAFLD in the offspring. This topic is very interesting and the results are promising. Nevertheless, the presentation of the data is of poor quality and the manuscript included various spelling and grammar mistakes. Authors have to highly improve the manuscript.
Major
- The abstract is very unspecific and has to be highly improved. What does terms such as “Inhibition of FXR” and “Increase inflammation and fatty acid synthase” mean? It could be expression, cellular location, activity?
- Authors stated to measure possible mechanism of NAFLD with the result of “related to the alterations in the gut microbiota and bile acid metabolism” without further specification. They did not investigate the mechanism of NAFLD development by high salt intake very well.
- Introduction: authors mix references of human studies and experiments in rodents without a specific description. It should be clearly stated, which organisms were investigated, especially in lines 56 ff. This is also relevant in the discussion section.
- The stated “Unique characteristic of NAFLD” were shown in Figure 1: histological microphotographs are of very poor quality. Figure 1G: Authors show one representative photo per group and describe a “greater periportal leukocyte infiltration” in the HS group. There are no differences in steatosis visible, no leukocytes at all visible in this quality and it is not clear, what is the main goal of the arrows. Figure 1H: again 1 representative photo per group. Here, steatosis is visible in higher amounts in A1 and B2, but again no quantification and no comparable photos. There might be a tendency of fibrosis in B1 and B2, but authors showed just two large portal fields in this group but only small PV (at least portal fields are not visible because of the poor photo quality) in A1 and A2. Authors should add quantitative analysis at least of steatosis. Furthermore, a grading of NAFLD by the NAS score from a pathologist is necessary. Quantification of leukocyte infiltration would be more obvious with F4/80 staining.
- Figure 2: Hepatic triglyceride synthesis is not primary regulated by expression of SREBP-1c, FAS and ACC but of their enzymatic activity that is posttranscriptional determined (e.g. cleavage, allosteric activation, (de)phosphorylation).
- Figure 4: cytokines should be measured on protein level.
- The order of the result presentation could be improved.
Minor
- Terms such as “High HS intake” are redundant
- Abbreviations (especially in Figures) should be explained
Author Response
Dear Dr. Reviewer,
I would like to thank you very much for your instructive letter on the manuscript revision. I want to express my deep thanks to the reviewers’ suggestions and comments to improve the paper quality.
According to all comments from the reviewer, we revised the manuscript carefully. The changes are red-colored in the manuscript and the detailed responses to the comments are listed as the following.
We hope that this revision can meet the journal requirements. Please let me know if further revision is required.
Best regards,
Wei Xia Ph.D.
1. The abstract is very unspecific and has to be highly improved. What does terms such as “Inhibition of FXR” and “Increase inflammation and fatty acid synthase” mean? It could be expression, cellular location, activity? Authors stated to measure possible mechanism of NAFLD with the result of “related to the alterations in the gut microbiota and bile acid metabolism” without further specification. They did not investigate the mechanism of NAFLD development by high salt intake very well.
Response: Thanks for the good suggestions. Accordingly, related revisions are made at line 18-26 of page 1. As follows “These alterations disturbed gut barrier by increased expression of tight junction protein (Tjp) and occludin (Ocln), and increased levels of systemic lipopolysaccharide (LPS) and hepatic inflammatory cytokine secretion (TNF-α and IL-6) in the liver. These changes inhibited expression of hepatic FXR signaling (CYP7A1 and FXR), thus triggering increased hepatic expression of pro-inflammatory cytokines (TNF-α and IL-6) and hepatic lipid metabolism associated genes (SREBP-1c, FAS, and ACC), and leading to unique characteristics of NAFLD. In conclusion, perinatal HS diet induced NAFLD of the weanling mice offspring, the possible mechanism was related to increased bacteria abundance of Proteobacteria and Bacteroides, LCA and DCA levels in feces, and expressions of hepatic FXR signaling.”
2. Introduction: authors mix references of human studies and experiments in rodents without a specific description. It should be clearly stated, which organisms were investigated, especially in lines 56 ff. This is also relevant in the discussion section.
Response: Thanks for the good suggestions. Accordingly, related revisions were made at line 37, 41-42,52-54, 57,61, 63 and 65 of page 2, respectively.
3. The stated “Unique characteristic of NAFLD” were shown in Figure 1: histological microphotographs are of very poor quality. Figure 1G: Authors show one representative photo per group and describe a “greater periportal leukocyte infiltration” in the HS group. There are no differences in steatosis visible, no leukocytes at all visible in this quality and it is not clear, what is the main goal of the arrows. Figure 1H: again 1 representative photo per group. Here, steatosis is visible in higher amounts in A1 and B2, but again no quantification and no comparable photos. There might be a tendency of fibrosis in B1 and B2, but authors showed just two large portal fields in this group but only small PV (at least portal fields are not visible because of the poor photo quality) in A1 and A2. Authors should add quantitative analysis at least of steatosis. Furthermore, a grading of NAFLD by the NAS score from a pathologist is necessary. Quantification of leukocyte infiltration would be more obvious with F4/80 staining.
Response: According to your suggestions, histological microphotographs new histological microphotographs have been taken and analyzed. H&E staining of liver tissue showed unique histologic features of NAFLD in the HS offspring, characterized by greater periportal leukocyte infiltration (Figure 1G). Black arrows indicate infiltrating lymphocytes. Masson staining of liver tissue showed slight fibrosis around the portal vein, while the fibrosis was not significantly different between the two groups (Figure 1H). The modified Pediatric NAFLD Histological Score (PNHS), which was used to for assessing histology change of NAFLD, was significantly higher in livers of the offspring with perinatal HS diet (Figure 1I).
4. Figure 2: Hepatic triglyceride synthesis is not primary regulated by expression of SREBP-1c, FAS and ACC but of their enzymatic activity that is posttranscriptional determined (e.g. cleavage, allosteric activation, (de)phosphorylation).
Response: We agree with the reviewer that hepatic triglyceride synthesis is not primary regulated by expression of SREBP-1c, FAS, and ACC, but of their enzymatic activity that is posttranscriptional determined (e.g. cleavage, allosteric activation, (de)phosphorylation). However, several studies have proved that overexpression of SREBP-1c, FAS, and ACC were associated with the accelerated hepatic lipogenesis in NAFLD animal model [45,46], while down-regulated mRNA expression of these genes can protect development of liver steatosis [47,48]. Therefore, up-regulated expression of SREBP-1c, FAS and ACC observed in our study suggested that perinatal HS diet induced hepatic lipogenesis and lipid accumulation.
5. Figure 4: cytokines should be measured on protein level.
The order of the result presentation could be improved.
Response: Thanks for the good suggestions. The cytokines on protein level were added accordingly in Figure 4 D-E. Also, the corresponding description were added in line 231 of page 8.
6. Terms such as “High HS intake” are redundant
Abbreviations (especially in Figures) should be explained
Response: Thanks. The “High” has been removed at lines 11 and 14 of page 1 and line 346 of page 11. All Abbreviations (especially in Figures) have been explained.

Round 2
Reviewer 1 Report
The authors did not provide sufficient evidence to respond to the reviewer's review.
Author Response
Dear Dr. Reviewer,
I would like to thank you very much for your instructive letter on the manuscript revision. I want to express my deep thanks to the reviewers’ suggestions and comments to improve the paper quality.
Considering that the last reply was not clear enough, here we revise the article again and revise the question. The changes are red-colored in the manuscript and the detailed responses to the comments are listed as the following.
We hope that this revision can meet the journal requirements. Please let me know if further revision is required.
Best regards,
Wei Xia Ph.D.
1. Authors demonstrated that perinatal HS lead to hepatic lipid accumulation in weanling offsprings. As shown in Fig. 1E, the hepatic TG levels were significantly increased by the HS diet. However,the HE and Masson staining results do not show a corresponding TG accumulation in the liver. Furthermore,although B1 and B2 in Fig. 1H were the results of two independent samples in the same group,the difference between B1 and B2 was obvious.
Response: As histological microphotographs are of poor quality, new histological microphotographs have been taken and analyzed. As shown in Fig. 1G-H, the HE and Masson staining results showed that steatosis are slightly higher in the offspring with perinatal HS diet. We have revised the results and the corresponding description were added in line 175-179 of page 3. Although histology of the liver did not show obvious lipid deformation, the biochemical indicators of pediatric NAFLD appeared, including hyperlipidemia. We are discussing this phenomenon in the discussion part and the corresponding description were added in line 266-277 of page 10. As follows“The reason for this phenomenon is the lack of accurate methods to identify liver steatosis. When the total area of steatosis is less than 20%, histologic of the liver to prove liver steatosis will be false positive [38]. Similarly, we did not find obvious TG accumulation from the results of HE and Masson staining liver tissue, but the levels of hepatic TG increased significantly. In addition,the modified Pediatric NAFLD Histological Score (PNHS) [30], which was used to for assessing histology change of NAFLD, was significantly higher in livers of the offspring with perinatal HS diet. Histological features were scored on a scale of 0-3 on steatosis, 0-3 on lobular inflammation, 0-2 on portal inflammation. Ballooning and fibrosis were not included in the scoring since they are characteristic features of NASH which was not seen in this model. The PNHS method has also been used in other animal experiments, and it has been proved that can identify NAFLD-like lesions in weaned mice. [22]”.
Fig. 1H show the structure of the portal vein was different between difference group. In order to see the histological microphotographs more clearly, two duplicate microphotographs were deleted one of them.
2. The gut microbiota results seem too simple,authors may consider adding the total bacteria levels and bar graph to display more detailed changes at phylum and family levels.
Response: The figures were added accordingly in Figure 3B-3E. Also, the corresponding description were added in lines 209-213 and of page 6.
3. Although microbiota changes were associated with tight junction disruption, whether dysbiosis plays a detrimental role in HF-induced tight junction disruption was still unclear. authors may consider using antibiotics or probiotics to prove this point.
Response: Thanks the reviewer for the great comments. Some studies have shown that HS intake can induce gut dysbiosis and disrupt gut barrier integrity, and antibiotics can restore HS intake-induced gut leakiness. Furthermore, antibiotic treatment can reverse the levels of Proteobacteria, Bacteroides, DCA, LCA, T TNF-α and IL-6 in mice. Consistent with these studies, we found that perinatal HS diet increased the abundance of Proteobacteria and Bacteroides, elevated offspring fecal LCA and DCA levels, and interfered bile acid homeostasis. Based on these reasons, we believe that gut dysbiosis is an important link that perinatal HS induced the in the progression of NAFLD of the offspring. However, due to scientific research funding issues, in this present study did not complete the antibiotic experiment. We added the above discussion to the discussion of the article and pointed out that future studies need to be confirmed by antibiotic experiments.
4. Hepatic proinflammatory response is associated with tight junction disruption. Authors should measure serum or plasma endotoxins levels in mice.
Response: Thanks for the good suggestions. The plasma endotoxins levels were added accordingly in Figure 4 C. Also, the corresponding description were added in lines 228-229 of page 8.
5. The results obtained by the author are independent phenomena, and the inner connection of these results should be discussed more.
Response: We agree with the reviewer. We added the inner connection of these results to the discussion of the article in lines 252-260 of page 10. As follows“We also found that perinatal HS diet increased the abundance of gut Proteobacteria and Bacteroides, and the concentration of DCA and LCA. These alterations disturbed gut barrier by increased expression of Tjp and Ocln [25], and increased levels of LPS and hepatic inflammatory cytokine secretion (TNF-α and IL-6) in the liver [34]. These changes inhibited expression of hepatic FXR signaling (CYP7A1and FXR), thus triggering increased hepatic expression of pro-inflammatory cytokines (TNF-α and IL-6) [35] and hepatic lipid metabolism associated genes (SREBP-1c, FAS, and ACC) [24], and leading to unique characteristics of NAFLD.”

Reviewer 2 Report
The authors have satisfied all my request,
Author Response
Thank you.
Reviewer 3 Report
The revised manuscript is highly improved. Authors include new analysis as recommended and present the data in a more specific and structured way.
Author Response
Thank you